# Beyond opioid prescribing: Evaluation of a substance use disorder curriculum for OBGYN residents

**Caitlin E. Martin**[1,2]*, **Bhushan Thakkar**[1], **Lauren Cox**[1], **Elisabeth Johnson**[3], **Hendrée E. Jones**[3], **AnnaMarie Connolly**[4]

**1** Department of Obstetrics and Gynecology, Virginia Commonwealth University School of Medicine, Richmond, VA, United States of America, **2** Institute of Drug and Alcohol Studies, Virginia Commonwealth University School of Medicine, Richmond, VA, United States of America, **3** Department of Obstetrics and Gynecology, UNC Horizons, University of North Carolina School of Medicine, Chapel Hill, NC, United States of America, **4** Department of Obstetrics and Gynecology, University of North Carolina School of Medicine, Chapel Hill, NC, United States of America

\* caitlin.martin@vcuhealth.org

## Abstract

**Data Availability Statement:** All relevant data are within the manuscript and its Supporting Information files.

### Objective

Amidst the current opioid crisis, there is a need for better integration of substance use disorder screening and treatment across specialties. However, there is no consensus regarding how to best instruct OBGYN trainees in the clinical skills related to opioid and other substance use disorders (SUD). Study objectives were (1) to assess the effectiveness a SUD curriculum to improve self-reported competence among OBGYN residents and (2) to explore its effectiveness to improve attending evaluations of residents' clinical skills as well as its feasibility and acceptability from the resident perspective.

### Methods

A pilot 3-session curriculum was developed and adapted to SUD screening and treatment which included readings, didactics, and supervised outpatient clinical experiences for OBGYN post-graduate year 1 (PGY-1) residents rotating through an integrated OBGYN-SUD clinic. Eighteen residents completed pre and post clinical skills self-assessments (SUD screening, counseling, referring, Motivational Interviewing) using an adapted Zwisch Rating Scale (range 1–5). Scores were compared between time points using paired t-tests. Sub-samples also (a) were evaluated by the attending on three relevant Accreditation Council for Graduate Medical Education Milestones (ACGME) milestone sets using the web-based feedback program, myTIPreport (n = 10) and (b) completed a qualitative interview (n = 4).

### Results

All PGY-1s (18/18) across three academic years completed the 3-session SUD curriculum. Clinical skill self-assessments improved significantly in all areas [SUD Screening (2.44 (0.98) vs 3.56 (0.62), p = <0.01); Counseling (1.81 (0.71) vs 3.56 (0.51), p = < .01; Referring (2.03 (0.74) vs 3.17 (0.71), p = < .01; Motivational Interviewing (1.94 (1.06) vs 3.33 (0.69), p

**Funding:** This study was supported by the Virginia Commonwealth University School of Medicine Educational Scholarship Fund. Dr. Martin is supported by NIDA award No. K23 DA053507 from the National Institute of Drug Abuse. This study was supported partially by UL1TR002649 and KL2TR002648 from the National Center for Advancing Translational Sciences. Drs. Jones and Johnson are supported by NIDA 1R01DA047867. The funders had no role in study design, data collection and analysis, decision to publish, or preparation of the manuscript.

**Competing interests:** The authors have declared that no competing interests exist.

= < .01)]. Milestone set levels assigned by attending evaluations (n = 10) also improved. Qualitative data (n = 4) revealed high acceptability; all curriculum components were viewed positively, and feedback was provided (e.g., desire for more patient exposures).

## Conclusion

A pilot SUD curriculum tailored for OBGYN PGY-1 residents that goes beyond opioid pre-scribing to encompass SUD management is feasible, acceptable and likely effective at improving SUD core clinical skills.

## Introduction

The current overdose crisis has shed light on the great burden of substance use disorders (SUD) across the nation [1]. To effectively combat the overdose crisis, a multimodal approach, supported by an array of stakeholders including policymakers, public health officials, clinical providers, and citizens themselves, is needed [2]. As consideration is given to the roles clinical providers can play, there has been a call for better integration of SUD screening, assessment and treatment across specialties [3]. In response to this call and the rising prevalence of opioid use disorder and its consequences, such as overdose [1], professional societies and health systems nationally have been leading efforts to improve access to training and resources for medical trainees [4]. Of note, however, is that most of these new educational opportunities have focused on opioid prescribing and have not emphasized the development of additional clinical skills related to substance use beyond such prescribing. This gap in training reflects the general response to the opioid crisis which, again, has focused primarily on opioid prescribing guidelines rather than on improving recovery-oriented SUD treatment services despite the importance of such services as outlined in the US Department of Health and Human Services' Five Point Strategy to Combat Opioid Abuse, Misuse and Overdose [2].

Surgical specialty residents are in an excellent clinical position to positively impact the opioid crisis given their significant roles in both postoperative pain management for patients receiving medication for opioid use disorder and preoperative care planning in collaboration with addiction medicine providers. Additionally, residents are exposed frequently to the complexities that surround inpatient care of people with SUD. Residents in primary care related specialties are also well positioned to apply learned SUD skills to their clinical practice as evaluation and management of chronic diseases other than addiction, such as diabetes and hypertension, are core to their training. Despite this clinical exposure, residency training in SUD prevention, screening, assessment, treatment and recovery is not as robust as warranted [5,6].

While the American College of Obstetricians and Gynecologists (ACOG) has started leading efforts to improve access to SUD training and resources [7] for OBGYN providers in practice [8], such training curricula do not exist specifically for OBGYN residents. Given the increasing prevalence of pregnancies affected by SUD [9,10] and rapid increase in opioid related overdose deaths in women over the past decade [11], OBGYN residents, given their clinical roles as both surgeons and primary care providers, are well positioned to effectively identify women with SUD as well as to care for women who use substances [12]. As such, residents could benefit from clinical training, across inpatient and outpatient settings, on education of SUD treatment that goes beyond opioid prescribing, as has been done for other diseases and conditions [13–15]. Such a multidisciplinary and multi-modal approach could better prepare residents to independently care for this unique patient population with complex

medical and biopsychosocial needs. Thus, the objectives of this study were (1) to evaluate the effectiveness of a novel pilot SUD curriculum created for PGY-1 OBGYN residents to improve resident self-reported competence in core substance use clinical skills and (2) to explore its ability to improve attending evaluations of residents' clinical skills according to the benchmarks set forth by the United States' guiding body for resident education, the Accreditation Council for Graduate Medical Education Milestones (ACGME), as well as its feasibility and acceptability from the resident perspective.

## Methods

### Study design

This was a prospective cohort study of PGY-1 OBGYN residents who completed a pilot curriculum directed towards competence in key clinical skills related to SUD prevention, screening, assessment, treatment and recovery. A curriculum was developed by a multidisciplinary team, including a psychologist and nurse practitioner with clinical experience in SUD during pregnancy, an OBGYN residency director with expertise in medical education, and an OBGYN resident. The curriculum was based on CREOG (Council on Resident Education in Obstetrics and Gynecology) objectives [16] and adapted to address key areas related to SUD (S1 Appendix). The curriculum was integrated into the Virginia Commonwealth University OBGYN PGY-1 ambulatory rotation starting in academic year 2018–2019. This study was approved by the Virginia Commonwealth University Institutional Review Board (HM20018443) and oral consent was obtained from all participants.

### Components of the pilot SUD curriculum

The pilot curriculum included 3 outpatient clinical sessions integrated into a specialized OBGYN clinic caring for women with SUD across the lifecourse, including pregnancy and postpartum. These 3 sessions focused on medical knowledge and clinical skills used to evaluate and manage patients with SUD. Additionally, each clinical session included an Attending-led didactic that focused on an assigned topic. The didactic was conducted in the clinical space for 15–30 minutes prior to first patient arrival (Table 1). Before each clinical session/didactic, residents completed required readings from ACOG committee opinions and the Substance Abuse and Mental Health Services Administration (SAMHSA) Clinical Guidance for the Treatment of Pregnant and Parenting Women with Opioid Use Disorder and their Infants [17]. Questions about the readings were reviewed with the attending during the didactic time.

**Table 1. Format of pilot curriculum on substance use disorders for OBGYN residents.**

| Clinical sessions | | Assigned topics | ACGME* Milestone Sets (evaluated in myTIPreport) |
|---|---|---|---|
| "Knows How" | "Shows How / Does" | | |
| 1 | 4 | Substance use disorder (SUD) definitions Pathophysiology of SUD and effects on pregnancy | **Patient Care—Office Practice:** Care of the patient with non-reproductive medical disorders **Professionalism**—Compassion, Integrity, and Respect for others |
| 2 | 5 | SBIRT (Screening, Brief Intervention, Referral, Treatment) for SUD | **Medical Knowledge**—Health Care Maintenance and Disease Prevention **Interpersonal and Communication Skills**—Communication with Patients and Families |
| 3 | 6 | SUD co-morbidities: medical, psychiatric, psychosocial and trauma history | **Medical Knowledge—Gynecology**: Perioperative care **Patient Care—Obstetrics:** Antepartum care and complications of pregnancy |

*ACGME- Accreditation Council for Graduate Medical Education.

Notably, starting in the second year (AY 2019–2020) of the curriculum, residents were scheduled to complete the pilot curriculum plus an additional 3 sessions, encompassing altogether an extended 6-session version of the curriculum. In doing so, sessions 1–3 of the pilot curriculum became termed the "knows-how" sessions based on Miller's pyramidal four-step hierarchical process [18]. During these "knows-how" sessions, residents were expected to demonstrate their knowledge as they saw patients and coordinated plans of care with the OBGYN-Addiction attending. Then the additional sessions 4–6 were termed "show how/does" sessions, again based on the Miller's pyramid "performance," where learners demonstrate, through their performance, that they are capable of using their knowledge while being supervised and observed [18]. While the topics of these "shows how/does" sessions (sessions 4–6) directly aligned with the topics of "knows-how" (sessions 1–3), the goals of these "shows-how/does" sessions were to (a) further strengthen clinical skills related to SUD (e.g., under attending supervision, the residents counsel patients regarding medications for opioid use disorder and provide directed feedback after proctoring residents practice Motivational Interviewing) and (b) practice integrating evidence-based strategies into patient care (e.g., pre-clinic readings included primary literature articles such as the MOTHER trial [19] rather than ACOG practice bulletins). The supervising attending then assessed whether learners were capable of functioning independently in these specific clinical situations.

## Study participants

Residents were informed of the new, required curriculum via email from the residency coordinator. The curriculum materials, including its objectives, readings and pre/post assessments, were put on the OBGYN residency's password protected website. In the first academic year of the curriculum, all PGY-1 residents were scheduled to complete the pilot curriculum in the spring semester. In the curriculum's second academic year onward, all six PGY-1 residents were scheduled to complete the pilot curriculum ('knows-how' sessions 1–3) in the fall semester, followed by the additional sessions ("shows-how/does" sessions 4–6) in the spring semester. Due to shifting of clinical responsibilities in the COVID-19 pandemic, only four out of the possible six residents completed all 6 sessions of the curriculum. These residents (n = 4) also completed both the self-assessments and interviews.

An overview of study participants and study assessments is provided in Fig 1. In brief, before and after the session sets, residents completed clinical skills self-assessments. Subsets of

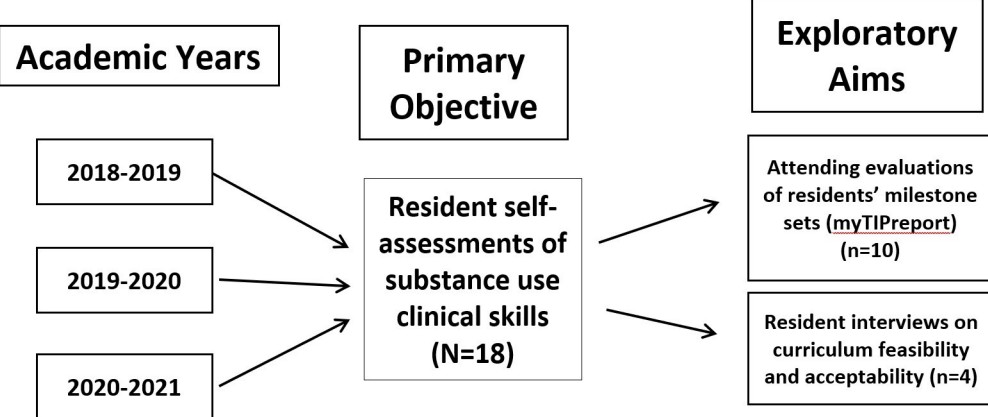

**Fig 1. Flow chart describing the study sample of PGY-1 OBGYN residents completing the pilot SUD curriculum and its evaluation assessments.**

the residents who completed all 'knows-how' and "shows-how/does" sessions were also invited to complete a semi-structured qualitative interview and to have their milestone set attending evaluations abstracted. These study assessments are detailed below.

## Primary objective: Curriculum effectiveness—clinical skills resident self-assessments

Before the first session of the pilot SUD curriculum, each resident completed a pre self-assessment on 4 core clinical skills related to SUD (pre-assessment). These core clinical skills were selected based on the Screening, Brief Intervention and Referral to Treatment (SBIRT) [20] model (Table 2). The self-assessment scale used was an adapted version of the Zwisch Rating Scale (range 1–5) [21] which asks the resident to self-report their needed supervision level needed regarding these 4 core clinical skills. After completion of the pilot curriculum (session #3), residents completed a post self-assessment on these same 4 clinical skills (post-assessment #1).

## Exploratory aim: Curriculum effectiveness—attending milestone evaluations using myTIPreport

For a sub-sample of residents completing the extended 6-session version of the pilot curriculum, (Fig 1), the OBGYN-Addiction attending physician completed milestone set feedback corresponding to each session's topic and clinical skills (Table 1) using the web-based program, myTIPreport [22,23]. myTIPreport was created as a cloud-based platform to provide "real-time" feedback on ACGME milestones and procedural skills. The ACGME is a private, not-for-profit organization that sets standards for US graduate medical education (residency and fellowship) programs and the institutions that sponsor them, and renders accreditation decisions based on compliance with these standards. ACGME Milestones provide a framework for the assessment of the development of the resident in key dimensions of the elements of physician competence in a specialty or subspecialty. A level was assigned to each milestone set assessed using ACGME milestone guidelines. The same milestone sets were evaluated after the "knows-how" and "shows-how/does" sessions, facilitating level comparisons across time.

## Exploratory aim: Curriculum feasibility and acceptability—semi-structured resident interviews

A subset of residents who completed the extended 6-session curriculum were also invited to participate in a semi-structured interview (Fig 1). The interview guide (S2 Appendix) was created to address the objectives of evaluating the feasibility and acceptability of the curriculum from the perspectives of the OBGYN residents. The questions addressed all the components of

**Table 2. Modified self-reported Zwisch rating scale for core clinical skills related to substance use disorders.**

| Show and Tell (Observation only) | Active Help (Direct supervision needed) | Passive Help (Indirect supervision needed) | Supervision Only (Supervision only for patient safety) | Expert (Can supervise others) |
|---|---|---|---|---|
| 1. What level of supervision do you think you need for screening patients for substance use disorders in the outpatient setting? | | | | |
| 2. What level of supervision do you think you need for counseling patients on the role substance use plays in their health? | | | | |
| 3. What level of supervision do you think you need for supporting and guiding patients using Motivational Interviewing techniques in changing their substance use behaviors? | | | | |
| 4. When indicated, what level of supervision do you think you need for referring patients for specialty treatment for their substance use? | | | | |

the curriculum as well as residents' perceptions of impact of the curriculum on their own clinical skills, self-learning and evidence-based practice. Additional questions focused on residents' responses on their self-assessments with the Zwisch scale and on suggestions for curriculum changes to further support data meeting the learning objectives. The final part of the interview queried on resident confidence in treating and caring for women with SUD and being an independent provider in the area of SUD. Interviews were conducted by a second year OBGYN resident (LC) and a research assistant (BT) trained in qualitative data collection. The interview setting was a private room for one resident and over videoconference for the other three residents due to COVID-19 restrictions. Interviews were audio-recorded with participant knowledge and consent. Interviews were semi-structured and designed to allow participants to tell their story in their own words. All 4 residents received a $20 gift card.

## Analysis

All self-assessments, milestone assessments, and audio-files were de-identified and labeled with a subject ID only. Paired t-tests were used to compare the global clinical skill self-assessment (range 1–5) between the pre and post assessments. Mean milestone set levels from the abstracted milestone set assessments, recorded in myTIPreport, are reported for the group over time. For the qualitative data, interviews were transcribed verbatim into Microsoft word. Data analysis occurred concurrently with data collection. The transcriptions were inductively analyzed using the "editing style" approach. The first step in this process involves "immersing" oneself in the participants' world to understand and interpret their experiences. This was accomplished by repeatedly re-reading the interviews and making notes in the margins, writing memos, and assigning codes in order to illuminate patterns and relationships that help to bring forth greater understanding of the data. The transcriptions were entered into Atlas.ti version 8 (Atlas.ti GmbH, Berlin, Germany). Coding was conducted by two investigators BT and LC who independently coded each interview. The two investigators adjudicated discrepancies in coding through discussion and modified codes as necessary to incorporate their interpretations with guidance from the principal investigator (CEM). Categories were later derived from the data themselves with input from the codes, and themes were created. Based on the codes, we calculated the frequency of responses. Quotations were selected that best identified responses to each theme.

## Results

### Primary objective: Curriculum effectiveness—clinical skills resident self-assessments

All 18 PGY-1 OBGYN residents across the three academic years participated in the SUD curriculum. The clinical skills assessment was completed by 18 OBGYN residents. As seen in Table 3, there was a significant improvement in resident self-reported supervision level needed

**Table 3. Resident clinical skills self-assessments before and after completion of the pilot SUD curriculum (N = 18).**

| | Pre Self-Assessment Mean (SD)* | Post Self-Assessment Mean (SD)* | p-value |
|---|---|---|---|
| Screening for substance use disorders | 2.44 (0.98) | 3.56 (0.62) | < 0.01 |
| Counseling patients with substance use disorders | 1.81 (0.71) | 3.56 (0.51) | < 0.01 |
| Use of Motivational Interviewing | 2.03 (0.74) | 3.17 (0.71) | <0.01 |
| Referring for addiction specialty treatment | 1.94 (1.06) | 3.33 (0.69) | < 0.01 |

*Mean and SD values of adapted Zwisch scale responses (range 1–5) for all participants.

to perform all corresponding clinical skills related to SUD after curriculum completion. As an example, for the core clinical skill where residents counseled patients with SUD, residents reported requiring, on average, direct supervision before the curriculum. After completing the curriculum, residents quantitatively reported improvement in this skill, stating they now required no more than indirect supervision. Similar improvements were observed for the other core clinical skills.

## Exploratory aim: Curriculum effectiveness—Attending milestone evaluations using myTIPreport

The milestone evaluations were completed by the teacher attending for 10 PGY-1 OBGYN residents. The attending assessments of milestone performance, abstracted from myTIPreport, demonstrated significant improvement across all milestone sets for the resident subset who completed the extended 6-session curriculum. The greatest improvement was seen for the Office Practice and Professionalism milestone sets. Table 4 describes the changes in the milestone sets from completion of the 'know-how' to the completion of the 'shows-how/does' sessions.

## Exploratory aim: Curriculum feasibility and acceptability—semi-structured resident interviews

The semi-structured interviews were completed by 4 PGY-1 OBGYN residents. The five major themes observed addressed: 1) Curriculum components positively contributing to improvements in evidence-based practice, self-directed learning and clinical skills development, 2) Insight into SUD clinical skills self-assessment results, 3) Impact of the curriculum experience on self-assessed SUD competence, 4) Opportunities to improve the curriculum, and 5) Barriers to improvement of SUD clinical skills, evidence-based practice and self-directed learning. Table 5 describes the themes and coding framework.

## Discussion

In the ongoing overdose crisis, there is a call nationally for providers and health systems across specialties to better integrate care for people with SUD [24]. However, OBGYN trainees are not receiving comprehensive education in SUD [25], along with residents in surgical [26] and primary care [27] specialties. We piloted a SUD curriculum for OBGYN residents to provide targeted SUD training beyond opioid prescribing. We found that integration of such a

**Table 4. Attending evaluations of milestone sets corresponding to SUD curriculum sessions.**

| ACGME milestone set assessments (as recorded in myTIPreport) (n = 4) | 'Knows-How' Level mean (SD) | 'Shows-How / Does" Level mean (SD) | p-value |
|---|---|---|---|
| **Patient Care—Office Practice**: Care of the patient with non-reproductive medical disorders | 2.15 (0.91) | 5.00 (0) | < 0.01 |
| **Professionalism**—Compassion, Integrity, and Respect for others | 2.00 (1.25) | 4.50 (0.82) | < 0.01 |
| **Medical Knowledge**—Health Care Maintenance and Disease Prevention | 1.25 (1.38) | 4.25 (1.51) | < 0.01 |
| **Interpersonal and Communication Skills**—Communication with Patients and Families | 1.95 (0.44) | 3.60 (1.05) | < 0.01 |
| **Medical Knowledge**—Perioperative care | 2.95 (0.55) | 3.95 (0.60) | < 0.01 |

*ACGME: Accreditation Council for Graduate Medical Education.

**Table 5. Resident interview themes with frequency counts and representative quotations on pilot SUD curriculum feasibility and acceptability (n = 4).**

| Major themes and sub-themes | n = Mentions (Participants) | Representative quotations |
|---|---|---|
| **Curriculum components positively contributing to improvements in SUD evidence-based practice, self-directed learning and clinical skills development** | **81 (4)** | |
| Clinical experiences working with patients with SUD | **21 (4)** | "What helped me the most, the second part was seeing the patients and having that feedback on how I was counseling was very important." |
| SUD clinical teaching and didactics | **11 (4)** | "At the beginning of the year it was like taking a full history. And then in the second half of the year, we worked more on like my counseling skills. So we'd pick one aspect of like, Oh, you can counsel on NAS or like, HEP C pregnancy or something." |
| SUD lectures during grand rounds | **2 (2)** | "I remember she gave her big grand rounds talk. Which I thought was really helpful and I happened to be in clinic. . .So it was really helpful cause it was kind of like putting it into use actively because I was on that rotation." |
| SUD readings and resources available on the OBGYN resident education blog | **6 (3)** | "Having just a really basic handout or booklet online to reference I thought. . .gave you kind of the tools to work with in her clinic, which was really invaluable." |
| SUD clinical skills real-time feedback | **10 (3)** | "The reading gave me a good background to start from, but I think watching her do it and having the real time feedback was probably the most helpful thing." |
| Self-directed SUD readings | **18 (4)** | "Reading the papers gave me a good background. So if the patients asked me questions, I felt confident like discussing the material with them. . .And I felt like that confidence allowed them to trust me more to tell [them] things." |
| **Insight into SUD clinical skills self-assessment results** | **60 (4)** | |
| Perceived limitations being a resident versus a fellow or attending | **5 (2)** | "I think I honestly feel pretty confident screening people, but it's because I'm a resident and I always, always need supervision. Like I need an attending." |
| Needing more practice with Motivational Interviewing | **8 (4)** | "I have been to many talks about motivational interviewing. I have read about it. I know a lot about it, but then when I try and put it in place, it's hard and I just need to keep practicing it more." |
| Reasons for self-assessment survey responses | **28 (4)** | "I mean, I feel competent asking patients the questions that you've been uncomfortable asking. And to become an expert. I think that would take a lot of experience [and] years of experience." |
| **Impact of curriculum experience on self-assessed competence in treating women with SUD** | **19 (4)** | |
| Confidence in referring patients for SUD treatment | **8 (4)** | "I feel comfortable within our own system knowing how things work and who to refer to." |
| Confidence in treating women with SUD after residency | **11 (4)** | "I think I feel pretty confident in screening women for substance use disorders, particularly if I was able to work it into a work-flow of a clinic." |
| **Opportunities to improve the SUD curriculum** | **73 (4)** | |
| General feedback on the SUD curriculum | **37 (4)** | "I definitely need more practice."<br>"And I don't know if it would actually make a difference of being able to actually go out into the community to another clinic outside of our own brick and mortar building." |
| More sessions and more SUD patient exposure | **31 (4)** | "Of course, the volume, doing it more often."<br>"I mean eight sessions is good. You get to be with her every week, but that's still, you know, at the end of the day that's only eight times that you've done it." |
| Tailoring the curriculum to individuals' unique learning styles | **5 (2)** | "I think it's just my learning style. I don't remember things as well when I just hear one lecture about it. You know, I have to do a little bit more to remember it." |
| **Barriers to improvements of SUD clinical skills, evidence-based practice and self-directed learning** | **5 (3)** | |

(*Continued*)

**Table 5.** (Continued)

| Major themes and sub-themes | n = Mentions (Participants) | Representative quotations |
|---|---|---|
| Resident time limitations | 5 (3) | "It was just like taking time to do the actual things sometimes when you're busy, even though the clinic and you think you should have a lot of time, sometimes it slips away from you, but I don't know" <br> "I think as far as the didactics go, I mean those are helpful but they don't always happen. Like depending on how busy the clinic day was." |

*SUD: Substance Use Disorders.

curriculum into residents' clinical teaching is feasible and acceptable to residents. Further, completion of this novel curriculum led to improvements in both trainees' self-reported supervision levels needed for key SUD clinical skills and attending milestone assessments of residents according to ACGME benchmarks in our small study sample.

OBGYN residents have the potential to play important roles in the optimization of SUD care across critical timeframes, including the perioperative [28] and perinatal periods [29]. It is imperative that trainees are comfortable and competent with identifying people with SUD. Both licit (i.e., tobacco, alcohol) and illicit (e.g., cocaine, non-prescribed opioids) substance use can significantly impact anesthesia effects intraoperatively and pain management postoperatively [30]. Recommendations are consistent that a well-devised, multi-disciplinary plan made preoperatively is ideal [30], including for people on medication for opioid use disorder (e.g., methadone, buprenorphine) [31]. During pregnancy, the same standards apply, as provision of SUD treatment within a comprehensive care model improves both maternal and fetal outcomes [32]. SUD that is not identified in either of these scenarios represents a missed opportunity to devise such care plans. Thus, OBGYN residents would likely benefit from effective training in substance use assessment. While medical schools are now more often teaching these skills [33], we found PGY-1 OBGYN residents self-reported needing significant supervision regarding clinical skills across the SBIRT model. After completion of the pilot curriculum, residents demonstrated significant improvement in self-reported supervision needed for these core SUD clinical skills, including screening, suggesting this curriculum could begin to address this critical gap in training.

After identification of substance use, OBGYN residents would further benefit from being able to provide brief interventions and link patients to SUD treatment when indicated. The perioperative and perinatal timeframes are windows of opportunity for patients to attain behavioral change for many reasons, such as how decreased substance use typically translates into decreased perioperative morbidity [34,35] and improved infant health outcomes [36]. Motivational Interviewing is an effective tool that providers can use to assist patients on this path [20]. At study baseline, residents reported the lowest levels of self-assessed competence in Motivational Interviewing across the four clinical skills assessed. In the pilot curriculum, a short didactic was given within the clinical setting with directed practice in Motivational Interviewing for tobacco use with patients. After curriculum completion, residents reported improved self-assessed competence in their Motivational Interviewing abilities. Notably, patients seen in the clinical setting of this study vary substantially in their stages of change, as the clinic prioritizes a low-threshold, harm reduction approach [37]; this represented a unique opportunity for residents to apply their new Motivational Interviewing skills and compassionate, person-centered approaches as role modeled by the clinic providers. Considering these findings and feedback from residents obtained in our qualitative interviews, future adaptations of this pilot SUD curriculum by other institutions should prioritize training in Motivational

Interviewing and harm reduction as it relates to substance use that includes both didactic and experiential components.

With the recent increased attention to opioid use disorder, most initial responses from providers, public health professionals and health systems across fields have centered on prevention–specifically decreasing opioid overprescribing [38]. While this response is encouraging, an effective approach to improving the care of people with SUD will likely require attention on more than just opioid prescribing. This aligns with the Department of Health and Human Services 5-Point Strategy to combat the opioid crisis [2]. In medical education, emerging innovative interventions show promise to prepare trainees in general [39,40], as well as residents in surgical specialties [4,41–44] to properly prescribe opioids. Yet approaches to provide residents with comprehensive SUD training are much more limited and, as such, needed [28,45]. Similar to prior work among primary care residents [46,47], this curriculum led to improvements in OBGYN residents' self-reported competence to link patients to and care for patients in SUD treatment, many of whom were on buprenorphine for opioid use disorder. An encouraging finding was that not only did residents' clinical skills self-assessments improve with the pilot curriculum, but also that attendings' assessments of residents showed significant improvements over time. Additionally, addiction stigma is, arguably, the most significant barrier to accessing and utilizing evidence-based SUD treatments [48], especially during pregnancy [49]. While not an outcome assessed in this study, stigma was addressed throughout the resident clinical experiences; for example, all residents are required to review addiction person-centered language recommendations [50] and are given immediate feedback on their language in the clinic. Since this study, we have added additional person-centered language training focused on addiction among pregnant and parenting people [51], with the goal to prepare the next generation of OBGYN providers to be leaders in destigmatizing the disease of addiction.

There is a significant gap between met and unmet SUD treatment need among women, including during pregnancy [52]. Equipping OBGYNs during residency in SUD evaluation and management is one avenue to close this critical gap in treatment access [29,53]. With this in mind, the pilot curriculum was designed to not only target key topics in SUD but also to address ACGME milestone assessments required of OBGYN residency training programs. The findings of both improvements in the ACGME milestone set levels assigned by attendings and high acceptability of the curriculum to residents as reported in the qualitative interviews highlight how SUD education does align well with the training environment of OBGYN residency. Overall, more research with larger sample sizes is needed in order to identify how to best provide proper education and training in SUD treatment for OBGYN residents and support residency program leadership in doing so.

Qualitatively, residents reported high acceptability and feasibility of the SUD curriculum. A motivating finding was how many residents stated they intend to integrate SUD management in some capacity into their future practice post-residency, and that they felt confident doing so. Most of the limitations of the curriculum discussed by residents revolved around their desire for more opportunities to apply their new knowledge and strengthen clinical skills, such as Motivational Interviewing, with directed teaching by the attending. While residents stated it was difficult at times to participate as fully as they intended due to time limitations and competing tasks, no residents stated the curriculum components were too burdensome. In response to resident feedback, we modified the curriculum for the 2020–2021 academic year with more varied resources to best match different learning styles (e.g., pre-recorded lecture [54]), opportunities for additional sessions during a 3rd year elective rotation, and an hour-long didactic on Motivational Interviewing for residents of all levels.

The strengths of this study lie in the innovative curriculum design. To our knowledge, no formalized curricula exist in the literature tailored to OBGYN residents focused on SUD

beyond opioid prescribing. Our approach was to design a pilot curriculum based on CREOG objectives that is also feasible within a busy outpatient clinic. This allowed us to direct the new curriculum to the specific needs and time limitations of OBGYN residents. The major weakness of this study is the small sample size and limited generalizability of the findings. In particular, only 4 residents completed the interviews and 10 had attending milestone assessments available for abstraction due to shifts in resident coverage in the COVID-19 pandemic. Despite these limitations, the results of the current study for this pilot curriculum informed the formation of a new, more comprehensive (e.g., additional components focused on destigmatizing SUD [50]), year-long curriculum that is currently being evaluated at our institution with plans to expand to other programs in the near future.

Residency directors want to improve trainee education to identify and care for people who use substances [26]. Providing trainees with the tools directed solely towards proper opioid prescribing is not sufficient to effectively combat the current opioid crisis. Further, given the stable national predominance of alcohol over drug use disorders among people with SUD [52] and increasing prevalence of methamphetamine misuse [9], residents need more robust SUD training not only in clinical skills beyond opioid prescribing but also in substances beyond opioids. As such, educational programming that allows residents to provide evidence-based care for people with SUD, including for people on medication for opioid use disorder, will be essential. OBGYN providers have unique opportunities to develop such clinically relevant educational programming given their interactions with patients throughout the perioperative and perinatal periods. Resident training programs will benefit from curricula that best prepare the next generation of OBGYN providers to improve the care of people with SUD. These findings highlight that integration of a SUD curriculum into a busy clinical environment is feasible, acceptable and likely effective at improving self-assessed competence in key clinical skills related to substance use.

## Supporting information

**S1 Appendix. CREOG objectives addressed by OBGYN residency pilot substance use disorder curriculum.**
(DOCX)

**S2 Appendix. Interview guide for OBGYN PGY-1s completing the substance use disorder pilot curriculum.**
(DOCX)

## Acknowledgments

The authors thank Dr. Sally Santen and the medical education office at the Virginia Commonwealth University School of Medicine for their guidance on publishing medical education and support of educational scholarship to Dr. Martin. We also thank Dr. Kate Miele for her assistance in providing input in the curriculum development. Lastly, we thank the residents of the OBGYN residency at the Virginia Commonwealth University for their willingness to participate in this pilot curriculum and provide such helpful feedback for continued work in this area.

## Author Contributions

**Conceptualization:** Caitlin E. Martin, Lauren Cox, Elisabeth Johnson, Hendrée E. Jones, AnnaMarie Connolly.

**Data curation:** Bhushan Thakkar, Lauren Cox, AnnaMarie Connolly.

**Formal analysis:** Bhushan Thakkar, Lauren Cox.

**Investigation:** Hendrée E. Jones, AnnaMarie Connolly.

**Methodology:** Caitlin E. Martin, Elisabeth Johnson, Hendrée E. Jones, AnnaMarie Connolly.

**Project administration:** Caitlin E. Martin, Bhushan Thakkar.

**Software:** Bhushan Thakkar, Lauren Cox.

**Supervision:** Caitlin E. Martin.

**Writing – original draft:** Caitlin E. Martin.

**Writing – review & editing:** Caitlin E. Martin, Bhushan Thakkar, Lauren Cox, Elisabeth Johnson, Hendrée E. Jones, AnnaMarie Connolly.

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
