## [Decision Letter · Decision Letter 0]

12 Aug 2022

PONE-D-22-16004Beyond Opioid Prescribing: Evaluation of a Substance Use Disorder Curriculum for OBGYN ResidentsPLOS ONE

Dear Dr. Martin,

Thank you for submitting your manuscript to PLOS ONE. After careful consideration, we feel that it has merit but does not fully meet PLOS ONE’s publication criteria as it currently stands. Therefore, we invite you to submit a revised version of the manuscript that addresses the points raised during the review process.

We look forward to receiving your revised manuscript.

Kind regards,

Nadine Harker, Ph.D

Academic Editor

PLOS ONE

Journal Requirements:

Reviewers' comments:

Reviewer's Responses to Questions

**Comments to the Author**

1. Is the manuscript technically sound, and do the data support the conclusions?

Reviewer #1: Yes

2. Has the statistical analysis been performed appropriately and rigorously? 

Reviewer #1: I Don't Know

3. Have the authors made all data underlying the findings in their manuscript fully available?

Reviewer #1: Yes

4. Is the manuscript presented in an intelligible fashion and written in standard English?

Reviewer #1: Yes

5. Review Comments to the Author

Reviewer #1: Abstract- some acronyms not explained and difficult to understand. Suggest removing those not explained if word count is an issue. (ACGME)

Ethical statement- What does IRB stand for?

Introduction-

Line 1- overdose crisis- opioid overdose or general overdose of all illicit substances?

Line 90- Non American readers will not know what ACGME is. Please provide a brief description/explanation of who they are and what they do.

Table 1: there is an * that explains ACGME but no corresponding *. this explanation should be earlier in text.

Methods

Line 138 notes 6 residents that were due to participate. Later in line 186 there are 4 interviews that take place. Cant follow any explanation of what happened to the other 2 and why not all participated.

Table 2 - the scale needs clarity. The scale is 1-4 but there are 5 descriptors (show how and Tell, Active Help, Passive Help, Supervision Only, Expert)

Results

Line 209- all 18 residents. This is confusing as initially there were 6, then 4 interviews and now 18 completed the curriculum. A better explanation of the numbers is needed.

Table 3 could be better interpreted after table 2 is clarified. It is clear there is an improvement in the PGs supervision needed but at what level and how do the scores correspond to the categories?

Line 221- what was the purpose of the extended curriculum? Who was it offered to and why was it not part of the original curriculum that was being studied?

Discussion

There is some room for discussion around stigma and advocacy for patients with SUD.

Some discussion around the concept of harm reduction would also be helpful as not all people are in the same stages of change (even with motivational interviewing) and preventing negative consequences of drug use should be seen as key to preventing esp infectious diseases like HIV and hepatitis.

General-

Good topic. i think its very important for specialists to be able to identify and manage SUDs.

The methods section and participant numbers and enrolment needs more clarity.

6. PLOS authors have the option to publish the peer review history of their article (what does this mean?). If published, this will include your full peer review and any attached files.

Reviewer #1: No

---

## [Author Response · Author response to Decision Letter 0]

19 Aug 2022

Journal Comments

Please provide additional details regarding participant consent. In the ethics statement in the Methods and online submission information, please ensure that you have specified what type you obtained (for instance, written or verbal, and if verbal, how it was documented and witnessed). If your study included minors, state whether you obtained consent from parents or guardians. If the need for consent was waived by the ethics committee, please include this information.

Thank you for your comments. We have included the consent details in Lines 104-105 and on the submission website. 

Reviewer Comments to the Author

Reviewer #1: Abstract- some acronyms not explained and difficult to understand. Suggest removing those not explained if word count is an issue. (ACGME)

Ethical statement- What does IRB stand for?

Our apologies for not explaining the acronyms. We have made the necessary changes throughout the manuscript. For example, in the abstract we have spelled out ACGME to “Accreditation Council for Graduate Medical Education Milestones.”

Introduction-

Line 1- overdose crisis- opioid overdose or general overdose of all illicit substances?

In line with the shift in drug use and overdose trends, we use the terminology “overdose crisis” as it refers to the epidemic of overdose deaths occurring due to opioids as well as other substances, such as methamphetamine (https://www.cdc.gov/drugoverdose/prevention/index.html). Our SUD curriculum, while it has a large focus on opioid use disorder, does also address intricacies of other use disorders. 

Line 90- Non American readers will not know what ACGME is. Please provide a brief description/explanation of who they are and what they do.

Table 1: there is an * that explains ACGME but no corresponding *. this explanation should be earlier in text.

We have added more information about the ACGME and its milestones in lines 89-91 and 170-175. For example, we now state in the introduction: “Thus, the objectives of this study were…to explore its ability to improve attending evaluations of residents’ clinical skills according to the benchmarks set forth by the United States’ guiding body for resident education, the Accreditation Council for Graduate Medical Education Milestones (ACGME)…” We also added a * to ACGME in Table 1 to correspond to the footnote for this acronym.

Methods

Line 138 notes 6 residents that were due to participate. Later in line 186 there are 4 interviews that take place. Cant follow any explanation of what happened to the other 2 and why not all participated.

Unfortunately, due to residents being pulled to cover clinical duties on other services at the height of the COVID-19 pandemic for our hospital, 2 residents were not able to complete all curriculum sessions in the academic year of recruitment for the qualitative interview. We have clarified this in the text: “Due to shifting of clinical responsibilities in the COVID-19 pandemic, only four out of the possible six residents completed all 6 sessions of the curriculum. These residents (n=4) also completed both the self-assessments and interviews.” We have included this in lines 142-144.

Table 2 - the scale needs clarity. The scale is 1-4 but there are 5 descriptors (show how and Tell, Active Help, Passive Help, Supervision Only, Expert)

Thank you for noting this typo. Our apologies for the confusion. The range of scores on the Zwisch Rating Scale is 1-5. We have made the changes in line number 159. 

Results

Line 209- all 18 residents. This is confusing as initially there were 6, then 4 interviews and now 18 completed the curriculum. A better explanation of the numbers is needed.

Thank you for your comments. Since our study was conducted across 3 academic years with the COVID-19 pandemic during that timeframe, our sample sizes for our outcomes corresponding to the study aims do differ. Figure 1 gives detail about our study sample. For clarity, for our three outcomes in the results text, we have also added a sentence at the start of every paragraph describing the number of residents who completed that outcome. 

For our primary objective assessing curriculum effectiveness, 18 PGY-1 OBGYN residents completed the Clinical skills resident self-assessments. This has been included in lines 216-217.

For our exploratory aim, where we assessed curriculum effectiveness using myTIP report, evaluations for 10 PGY-1 OBGYN residents were completed by the teacher Attending. This has been included in lines 231-232.

For our exploratory aim, where we assessed curriculum feasibility and acceptability using semi-structured resident interviews, 4 PGY-1 OBGYN residents completed the interview. This has been included in line 242.

Table 3 could be better interpreted after table 2 is clarified. It is clear there is an improvement in the PGs supervision needed but at what level and how do the scores correspond to the categories?

We have added more information explaining the results in lines 219-224: “As an example, for the core clinical skill where residents counseled patients with SUD, residents reported requiring, on average, direct supervision before the curriculum. After completing the curriculum, residents quantitatively reported improvement in this skill, stating they now required no more than indirect supervision. Similar improvements were observed for the other core clinical skills.”

Line 221- what was the purpose of the extended curriculum? Who was it offered to and why was it not part of the original curriculum that was being studied?

For the academic year 2018-2019, when the pilot curriculum was initiated, the curriculum included 3 outpatient clinical sessions; this was primarily driven by what accommodated the resident schedules and the fact that the OBGYN-Addiction faculty leading the curriculum joined the institution that academic year. During these 3 sessions, residents were expected to demonstrate their knowledge as they saw patients and coordinated plans of care with the OBGYN-Addiction attending. After reviewing the results from the first year and positive feedback from residents, the OBGYN-Addiction faculty added three more sessions to the curriculum, and the residency leadership accommodated resident schedules to complete the extended curriculum in academic year 2019-2020. In these additional sessions, the residents were expected to demonstrate, through their performance, that they are capable of using their knowledge gained during the initial three sessions while being supervised and observed. The 6 session (extended curriculum) continued to be offered to residents during the third academic year of this study, 2020-2021. 

Discussion

There is some room for discussion around stigma and advocacy for patients with SUD.

Some discussion around the concept of harm reduction would also be helpful as not all people are in the same stages of change (even with motivational interviewing) and preventing negative consequences of drug use should be seen as key to preventing esp infectious diseases like HIV and hepatitis.

We agree; these topics are very important! 

We have added more discussion about harm reduction with a citation about our clinic to lines 289-293: “Notably, patients seen in the clinical setting of this study vary substantially in their stages of change, as the clinic prioritizes a low-threshold, harm reduction approach; this represented a unique opportunity for residents to apply their new Motivational Interviewing skills and compassionate, person-centered approaches as role modeled by the clinic providers.” 

We have added more discussion about stigma with citations to lines 311-318: “Additionally, addiction stigma is, arguably, the most significant barrier to accessing and utilizing evidence-based SUD treatments, especially during pregnancy. While not an outcome assessed in this study, stigma was addressed throughout the resident clinical experiences; for example, all residents are required to review addiction person-centered language recommendations and are given immediate feedback on their language in the clinic. Since this study, we have added additional person-centered language training focused on addiction among pregnant and parenting people, with the goal to prepare the next generation of OBGYN providers to be leaders in destigmatizing the disease of addiction.”

General-

Good topic. i think its very important for specialists to be able to identify and manage SUDs.

The methods section and participant numbers and enrolment needs more clarity.

Thank you for the feedback; we appreciate it. We believe we have addressed the comments above adequately with additional clarity about our study design and sample to couple Figure 1.

---

## [Editor Report · Decision Letter 1]

31 Aug 2022

Beyond Opioid Prescribing: Evaluation of a Substance Use Disorder Curriculum for OBGYN Residents

PONE-D-22-16004R1

Dear Dr. Martin,

We’re pleased to inform you that your manuscript has been judged scientifically suitable for publication and will be formally accepted for publication once it meets all outstanding technical requirements.

Kind regards,

Nadine Harker, Ph.D

Academic Editor

PLOS ONE
---

## [Editor Report · Acceptance letter]

6 Sep 2022

PONE-D-22-16004R1 

Beyond Opioid Prescribing: Evaluation of a Substance Use Disorder Curriculum for OBGYN Residents 

Dear Dr. Martin:

I'm pleased to inform you that your manuscript has been deemed suitable for publication in PLOS ONE. Congratulations! Your manuscript is now with our production department. 

Kind regards, 

on behalf of

Prof Nadine Harker 

Academic Editor

PLOS ONE